# Serially assessed bisphenol A and phthalate exposure and association with kidney function in children with chronic kidney disease in the US and Canada: A longitudinal cohort study

Melanie H. Jacobson[1]☯*, Yinxiang Wu[2]☯, Mengling Liu[2,3], Teresa M. Attina[1], Mrudula Naidu[1], Rajendiran Karthikraj[4,5], Kurunthachalam Kannan[1,4,5], Bradley A. Warady[6], Susan Furth[7], Suzanne Vento[8], Howard Trachtman[8], Leonardo Trasande[1,2,3,9,10]

1 Division of Environmental Pediatrics, Department of Pediatrics, NYU Langone Medical Center, New York, New York, United States of America, 2 Department of Population Health, NYU Langone Medical Center, New York, New York, United States of America, 3 Department of Environmental Medicine, NYU Langone Medical Center, New York, New York, United States of America, 4 Wadsworth Center, New York State Department of Health, Albany, New York, United States of America, 5 Department of Environmental Health Sciences, School of Public Health, State University of New York at Albany, Albany, New York, United States of America, 6 Division of Nephrology, Department of Pediatrics, Children's Mercy Kansas City, Kansas City, Missouri, United States of America, 7 Division of Nephrology, Department of Pediatrics, Children's Hospital of Philadelphia, Philadelphia, Pennsylvania, United States of America, 8 Division of Nephrology, Department of Pediatrics, NYU Langone Medical Center, New York, New York, United States of America, 9 Wagner Graduate School of Public Service, New York University, New York, New York, United States of America, 10 School of Global Public Health, New York University, New York, New York, United States of America

☯ These authors contributed equally to this work.
* melanie.jacobson2@nyulangone.org

**Data Availability Statement:** The data underlying the results presented in the study are available

## Abstract

### Background

Exposure to environmental chemicals may be a modifiable risk factor for progression of chronic kidney disease (CKD). The purpose of this study was to examine the impact of serially assessed exposure to bisphenol A (BPA) and phthalates on measures of kidney function, tubular injury, and oxidative stress over time in a cohort of children with CKD.

### Methods and findings

Samples were collected between 2005 and 2015 from 618 children and adolescents enrolled in the Chronic Kidney Disease in Children study, an observational cohort study of pediatric CKD patients from the US and Canada. Most study participants were male (63.8%) and white (58.3%), and participants had a median age of 11.0 years (interquartile range 7.6 to 14.6) at the baseline visit. In urine samples collected serially over an average of 3.0 years (standard deviation [SD] 1.6), concentrations of BPA, phthalic acid (PA), and phthalate metabolites were measured as well as biomarkers of tubular injury (kidney injury molecule-1 [KIM-1] and neutrophil gelatinase-associated lipocalin [NGAL]) and oxidative

from the Chronic Kidney Disease in Children Study (https://repository.niddk.nih.gov/studies/ckid/). The data are owned by a third party and the authors do not have permission to share the data. This study represents the work of an ancillary study to the parent CKiD study. Access to the data may be arranged through contacting the parent study through the given link.

**Funding:** This work was supported by the National Institutes of Health, National Institute of Diabetes and Digestive and Kidney Diseases (https://www.nih.gov/about-nih/what-we-do/nih-almanac/national-institute-diabetes-digestive-kidney-diseases-niddk) grant number R01 DK100307 (LT and HT). The funders had no role in study design, data collection and analysis, decision to publish, or preparation of the manuscript.

**Competing interests:** I have read the journal's policy and the authors of this manuscript have the following competing interests: HT reports NIDDK grants and consultancy agreements with Retrophin, Goldfinch Bio, Chemocentryx and Otsuka Pharmaceutical.

**Abbreviations:** BPA, bisphenol A; CKD, chronic kidney disease; CKiD, Chronic Kidney Disease in Children; CV, coefficient of variation; DBP, diastolic blood pressure; DEHP, di(2-ethylhexyl) phthalate; DOP, di-n-octyl phthalate; eGFR, estimated glomerular filtration rate; ESKD, end-stage kidney disease; HMW, high molecular weight; LME, linear mixed-effects; LMW, low molecular weight; LOD, limit of detection; PA, phthalic acid; SBP, systolic blood pressure; UPCR, urinary protein-to-creatinine ratio.

stress (8-hydroxy-2′-deoxyguanosine [8-OHdG] and $F_2$-isoprostane). Clinical renal function measures included estimated glomerular filtration rate (eGFR), proteinuria, and blood pressure. Linear mixed models were fit to estimate the associations between urinary concentrations of 6 chemical exposure measures (i.e., BPA, PA, and 4 phthalate metabolite groups) and clinical renal outcomes and urinary concentrations of KIM-1, NGAL, 8-OHdG, and $F_2$-isoprostane controlling for sex, age, race/ethnicity, glomerular status, birth weight, premature birth, angiotensin-converting enzyme inhibitor use, angiotensin receptor blocker use, BMI $z$-score for age and sex, and urinary creatinine. Urinary concentrations of BPA, PA, and phthalate metabolites were positively associated with urinary KIM-1, NGAL, 8-OHdG, and $F_2$-isoprostane levels over time. For example, a 1-SD increase in ∑di-n-octyl phthalate metabolites was associated with increases in NGAL (β = 0.13 [95% CI: 0.05, 0.21], $p = 0.001$), KIM-1 (β = 0.30 [95% CI: 0.21, 0.40], $p < 0.001$), 8-OHdG (β = 0.10 [95% CI: 0.06, 0.13], $p < 0.001$), and $F_2$-isoprostane (β = 0.13 [95% CI: 0.01, 0.25], $p = 0.04$) over time. BPA and phthalate metabolites were not associated with eGFR, proteinuria, or blood pressure, but PA was associated with lower eGFR over time. For a 1-SD increase in ln-transformed PA, there was an average decrease in eGFR of 0.38 ml/min/1.73 m$^2$ (95% CI: −0.75, −0.01; $p = 0.04$). Limitations of this study included utilization of spot urine samples for exposure assessment of non-persistent compounds and lack of specific information on potential sources of exposure.

## Conclusions

Although BPA and phthalate metabolites were not associated with clinical renal endpoints such as eGFR or proteinuria, there was a consistent pattern of increased tubular injury and oxidative stress over time, which have been shown to affect renal function in the long term. This raises concerns about the potential for clinically significant changes in renal function in relation to exposure to common environmental toxicants at current levels.

## Author summary

### Why was this study done?

- The prevalence of chronic kidney disease has been steadily increasing over the last 40 years.

- While there are several known risk factors, such as diabetes and hypertension, few are potentially modifiable.

- Recent work has suggested kidney function may be affected by exposure to environmental chemicals such as bisphenols and phthalates, which are synthetic compounds used in the manufacturing of plastics and other consumer products.

- However, these chemicals are non-persistent, and no longitudinal studies to our knowledge have been conducted to investigate their potential impact on kidney function over time.

### What did the researchers do and find?

- In urine samples collected annually over 5 years from 618 children and adolescents with chronic kidney disease, we measured bisphenol A, phthalates, and biomarkers of tubular injury and oxidative stress.

- Clinical renal function outcomes were also monitored and collected over 5 years.

- We found that bisphenol A and phthalates were associated with increased tubular injury and oxidative stress biomarkers.

- Although phthalic acid was associated with lower estimated glomerular filtration rate over time, neither bisphenol A nor other phthalates were associated with clinical renal outcomes.

### What do these findings mean?

- Bisphenol A and phthalates were not associated with clinical renal function outcomes, but showed consistent positive associations with tubular injury and oxidative stress, which may signal the potential for clinical events to manifest with prolonged follow-up.

- These findings raise concern about the potential for clinically relevant changes to renal function to develop over time in relation to environmental exposures at current levels.

- This study suggests that exposure to environmental chemicals may be a potentially modifiable risk factor for the progression of chronic kidney disease.

## Introduction

Chronic kidney disease (CKD) is a growing health problem in children and adults [1]. In the United States between 2002 and 2016, there was a significant increase in disability-adjusted life years and mortality related to CKD, especially in young adults [2]. Furthermore, the incidence and prevalence of CKD among children has been steadily increasing since the 1980s [3–6]. CKD can progress to end-stage kidney disease (ESKD), which requires either renal replacement therapy (such as dialysis) or transplantation, which has significant impacts on morbidity, mortality, and healthcare costs [7–9].

There are a number of traditional risk factors associated with CKD progression to ESKD in children including hypertension, obesity, diabetes, and altered divalent mineral metabolism [3,10–12]. In addition, there is growing evidence suggesting that exposure to environmental chemicals may play a role as well [13]. For example, heavy metals such as lead and arsenic have been established as environmental nephrotoxins [14]. Furthermore, several recent studies have documented associations between other more common and low-level synthetic chemical exposures, such as bisphenol A (BPA) and phthalates, and kidney function [15–17]. These exposures have been hypothesized to affect kidney function through promotion of oxidative stress [18–24]. Because it has been shown that the kidney is vulnerable to environmental toxicants, especially prenatally and in early life [14,25], these exposures may be particularly relevant to pediatric CKD.

Surveys of healthy children and adults in the US indicate that exposure to BPA and phthalates is ubiquitous [26–28], and children are disproportionately exposed [29,30]. BPA is used in

the manufacturing of polycarbonate plastics, medical devices, metal food can sealants, and thermal paper receipts [31,32], and phthalates are found in personal hygiene and cosmetic products [33,34], food packaging, medical devices (e.g., in polyvinyl chloride [PVC] intravenous tubing and in medications), and construction materials [35,36]. The often unavoidable sources of exposure beginning in childhood and adolescence may lead to a greater cumulative toxicological burden over a lifetime [37,38]. This may be exacerbated in children with CKD because a reduced glomerular filtration rate and impaired tubular secretion may compromise renal excretion of BPA and phthalates [39]. In addition, sources of exposure may be greater due to medication use and frequent contact with plastics in medical devices and tubing, including dialyzer materials [40,41].

We previously reported findings from a cross-sectional analysis of urinary BPA and phthalate levels in relation to kidney function in children and adolescents with CKD enrolled in the Chronic Kidney Disease in Children (CKiD) observational study [16,42]. Our results demonstrated lower urinary excretion of BPA and phthalates in children with CKD compared with the general healthy pediatric population. Urinary BPA levels were not significantly associated with any index of kidney function, but low molecular weight phthalates were associated with lower proteinuria and greater estimated glomerular filtration rate (eGFR). However, pharmacokinetic studies have shown that BPA and phthalates are rapidly metabolized and have half-lives of less than 24 hours in the human body [43–46]. Studies with serial urinary BPA measurements have identified correlation coefficients in the range of 0.22–0.57 over 1- to 6-month periods [47,48], which suggests that spot measurements at single time points may not be sufficient to characterize chronic exposure. Furthermore, any effects of BPA and phthalates on kidney function are likely to be cumulative. Therefore, a more complete investigation of the potential adverse effects of exposure to these compounds may require serial measures over time assessed in relation to the longitudinal course of kidney function.

In this study, we evaluated the associations between longitudinally measured urinary BPA and phthalates and the trajectory of renal function over a 5-year observation period using a variety of measures over time: eGFR, proteinuria, systolic and diastolic blood pressure (SBP and DBP, respectively), and urinary biomarkers of tubular injury: kidney injury molecule-1 (KIM-1) and neutrophil gelatinase-associated lipocalin (NGAL). In addition, in order to assess the plausibility of a potential mechanism of action, we examined the associations between longitudinally measured urinary BPA and phthalates and serially assessed urinary oxidative stress biomarkers: 8-hydroxy-2′-deoxyguanosine (8-OHdG) and $F_2$-isoprostane.

## Methods

A prospective analysis plan was followed and is provided in S1 Text. This study is reported as per the Strengthening the Reporting of Observational Studies in Epidemiology (STROBE) guideline (S1 STROBE Checklist).

### Study population

The CKiD study is a multi-center prospective cohort study of children aged 6 months to 16 years with mild-to-moderate CKD with the overall goal of identifying predictors and sequelae of CKD progression. The CKiD study procedures and protocol have been previously described [42,49]. Briefly, pediatric patients with prevalent CKD were recruited at 57 clinical sites throughout the US and Canada, such as Boston, Massachusetts; Kansas City, Missouri; Seattle, Washington; and Winnipeg, Manitoba [42]. Children were enrolled, assessed 3–6 months after study entry (i.e., baseline), and every year thereafter. These annual data collection study visits were conducted until the initiation of renal replacement therapy for treatment of ESKD.

Annual study visits included a physical examination conducted by study staff, biological specimen collection (e.g., urine samples; specimens were stored in a biorepository for future use in ancillary studies), standardized blood pressure measurement, and questionnaire administration. Biological specimens used in this study were collected between July 21, 2005, and November 12, 2015, and were stored at $-80$ ºC in a central biorepository until being used for exposure and outcome assessment.

## Ethics statement

The institutional review board at each CKiD study site approved the study protocol, and all research was performed in accordance with established guidelines. Written informed consent was obtained from all parents or legal guardians, and assent from all participants depending on their age and institutional guidelines. The New York University School of Medicine Institutional Review Board deemed this project exempt from review due to data collection being complete and the dataset de-identified.

## Exposure measures

Longitudinally collected and stored urine samples were used for exposure assessment. Phthalic acid (PA), 21 individual phthalate metabolites, BPA, and creatinine were analyzed at the Wadsworth Center, New York State Department of Health, Albany, NY. Details on the analytic methods for the phthalates and BPA have been previously described [16]. Briefly, urine samples were processed using enzymatic deconjugation and extracted, and analysis was performed with high-performance liquid chromatography coupled with electrospray tandem mass spectrometry under negative mode of ionization.

For all exposures, measures below the limit of detection (LOD) were imputed by the LOD divided by the square root of 2 [50]. For statistical analyses, phthalates were analyzed in groups as micromolar sums of metabolites corresponding to their use in product categories: Σdi (2-ethylhexyl) phthalate (DEHP), Σhigh molecular weight (HMW) phthalates (comprising metabolites $\geq$ 250 Da), Σlow molecular weight (LMW) phthalates (comprising metabolites < 250 Da), and Σdi-n-octyl phthalate (DOP). All calculations excluded metabolites that were detected in <50% of samples (i.e., mCHP, mOP, mIDP, mPeP, mIPrP, and mINP). Thus, sums were calculated with the following metabolites: ΣDEHP: mECPP, mEHHP, mEOHP, and mCMHP; ΣHMW phthalates: mCPP, mECPP, mEHHP, mEOHP, mBzP, mHxP, mHPP, mCMHP, mCHpP, mCIOP, and mCINP; ΣLMW phthalates: mMP, mEP, mBP, and mIBP; and ΣDOP: mCPP and mCHpP. BPA and PA were examined as individual exposures.

## Outcomes

**Renal function measures.** Several correlates of renal function were examined over time. The primary outcome was eGFR, calculated using the modified Schwartz (i.e., bedside) equation [51]: eGFR (ml/min/1.73 m$^2$) = 0.413 × height (cm)/serum creatinine.

Other outcome measures of interest included urinary protein-to-creatinine ratio (UPCR), SBP, and DBP. Analytical methods for these measures in the CKiD study have been previously described. All laboratory measures were conducted at the central CKiD laboratory (University of Rochester) [52,53]. Briefly, total urine protein was determined using an immunoturbidimetric assay, and UPCR was calculated as the ratio of total urinary protein concentration to urinary creatinine concentration (mg/dl:mg/dl). SBP and DBP were measured in the right arm by auscultation using an aneroid sphygmomanometer [54]. Three measurements at 30-second intervals were taken, and the average of the 3 readings was calculated. Blood pressure measures

were standardized to *z*-scores according to the National High Blood Pressure Education Program Fourth Report [55].

**Biomarkers of tubular injury.** Two biomarkers of tubular injury were measured: kidney injury molecule-1 (KIM-1) and neutrophil gelatinase-associated lipocalin (NGAL). KIM-1 and NGAL were measured in urine samples at the New York University High Throughput Biology Laboratory. They were quantified by solid phase sandwich ELISAs using the Quantikine Human TIM-1 Immunoassay and Quantikine Human Lipocalin-2 Immunoassay, respectively (R&D Systems, Minneapolis, MN), according to the manufacturer protocol. All analyses were conducted in duplicate. Intra-assay coefficients of variation (CVs) ranged from 3.6% to 3.7% for KIM-1 and from 2.3% to 3.9% for NGAL. Inter-assay CVs ranged from 0.7% to 4.3% for KIM-1 and from 0.6% to 4.8% for NGAL. Measures below the LOD were imputed by the LOD divided by the square root of 2 [50].

**Oxidative stress.** Urine samples with sufficient volume were analyzed for 8-OHdG and $F_2$-isoprostane at the New York University High Throughput Biology Laboratory ($N$ = 2,465 samples, $N$ = 618 individuals, and $N$ = 1,287 samples, $N$ = 522 individuals, respectively). 8-OHdG was quantified by competitive ELISA using the OxiSelect Oxidative DNA Damage ELISA (Cell Biolabs, San Diego, CA). Similarly, $F_2$-isoprostane was measured with a competitive enzyme-linked immunoassay, the OxiSelect 8-iso-Prostaglandin $F_{2}\alpha$ ELISA Kit (Cell Biolabs). All analyses were conducted in duplicate as directed by the manufacturer. Intra-assay CVs ranged from 4.6% to 11.1% for 8-OHdG and from 6.8% to 9.7% for $F_2$-isoprostane. Inter-assay CVs ranged from 3.9% to 16.6% for 8-OHdG and from 14.2% to 15.9% for $F_2$-isoprostane. Measures below the LOD were imputed by the LOD divided by the square root of 2 [50].

## Statistical analysis

Participant characteristics were evaluated over time. Counts and percentages were calculated for categorical variables, and means and standard deviations (SDs) (or medians and interquartile ranges [IQRs], as appropriate) are presented for continuous variables. Distributions of all environmental exposures, renal outcomes, and biomarkers of tubular injury and oxidative stress were examined by study visit over time. Due to the right-skewed distributions of all urinary biomarkers of environmental exposures, they were natural-log-transformed in order to reduce the influence of extreme outliers. Biomarkers of tubular injury, KIM-1 and NGAL; oxidative stress biomarkers, 8-OHdG and $F_2$-isoprostane; and UPCR were also natural-log-transformed before multivariable analysis in order to better approximate a normal distribution.

In order to estimate the longitudinal associations between serially assessed BPA and phthalate measures and the longitudinal course of the outcomes of interest (i.e., eGFR, UPCR, SBP, DBP, KIM-1, NGAL, 8-OHdG, and $F_2$-isoprostane), separate linear mixed-effects (LME) models were fit for each exposure–outcome combination. Random intercepts were included to account for cross-subject heterogeneity and within-individual correlation. In addition, based on both graphic assessments of between-visit correlations of the outcomes and a likelihood ratio test, a first-order autoregressive (AR(1)) covariance structure was applied in order to model the correlation pattern over time.

For each outcome, we first fit a LME model with log-transformed exposure variable, time (i.e., months from the baseline assessment), and a quadratic term for time included as main fixed effects, as well as baseline covariates hypothesized to act as confounders and/or key predictors of the outcome based on previous literature [16,17,42,56]. These included sex, age, race/ethnicity, glomerular status, birth weight, premature birth, angiotensin-converting enzyme inhibitor (ACEI) use, angiotensin receptor blocker (ARB) use, BMI *z*-score for age and sex, and time-dependent log-transformed creatinine. Models were fit with chemical

exposures expressed on a volume basis (ng/ml or nmol/ml), and creatinine was controlled for as a covariate instead of indexing the exposure measures by creatinine (ng/mg Cr or nmol/mg Cr), in order to separate any impacts of the chemicals and creatinine [57], and because the same urinary creatinine measure was included in one of the primary outcomes of interest (i.e., UPCR), which could induce a spurious correlation due to urinary creatinine associations rather than chemical exposure associations [39]. In models for eGFR, UPCR, KIM-1, NGAL, 8-OHdG, and $F_2$-isoprostane, SBP $z$-score was additionally controlled for. In all models, the log-transformed exposure variables were standardized so that the estimates represented the average change in the outcome in response to a 1-SD change in the log-transformed exposure.

From these models, if there was a statistically significant ($p < 0.05$) estimate for the main effect of the exposure variable, we further examined the potential for an interaction between the exposure variable (centered with mean = 0) and time. In subsequent models, interaction cross-product terms between each exposure and time (both linear and quadratic terms) were included, and time-specific estimates of exposure were output and plotted by linear contrasts from the fitted models. Details of the modeling strategy are provided in S1 Fig.

Finally, we conducted sensitivity analyses. First, we evaluated the potential for associations of the outcomes with cumulative levels of BPA and phthalate metabolites over time. The cumulative average over time was calculated as the average of the log-transformed measures up until each visit time (i.e., cumulative average). These measures were then fitted using the same LME modeling strategy as described above. Second, we examined whether controlling for year of study enrollment affected the measures of association.

Analyses were performed using complete case analysis. All analyses were conducted in R version 3.5.0 for Windows [58], using the package "nlme" [59] version 3.1–137 for LME models.

## Results

The analytic sample included 618 participants, contributing 2,469 total visits (mean number of visits ± SD: 4.0 ± 1.6), representing 1,839.3 total person-years of observation (3.0 ± 1.6 years per patient). Most study participants were male (63.8%) and white (58.3%), and participants had a median age of 11.0 years (IQR = 7.6 to 14.6) at the baseline visit (Table 1). The majority (89.2% overall and 88.3% at baseline) of the patients had non-glomerular disease. At baseline, kidney function was moderately impaired, with an average eGFR of 51.9 ml/min/1.73 m$^2$ (standard deviation = 19.8) and most participants (60.4%) having stage G3a CKD or better ($\geq$45 ml/min/1.73 m$^2$). At baseline, participants had on average slightly above average body mass index (BMI) (mean ± SD $z$-score adjusted for age and sex: 0.4 ± 1.2) and SBP (mean ± SD $z$-score adjusted for age, sex, and height: 0.3 ± 1.1), which both generally decreased over the course of follow-up.

BPA was detected in 78.3% of samples (Table 2). PA was detected in 99.1% of samples, and detection frequencies for the individual phthalate metabolites included in the group sums ranged from 50% to 100%. The distributions of urinary BPA and phthalates were right-skewed, with the 95th percentile typically being an order of magnitude greater than the median level. Urinary concentrations of exposures had a substantial degree of within-individual variability over time, with intraclass correlation coefficients ranging from 0.12 for ΣDOP to 0.39 for ΣLMW phthalates (S1 Table). Over the course of the study enrollment period (i.e., 2005–2015), chemical exposure concentrations decreased over time, although within-individual concentrations of some chemicals remained steady or increased slightly over follow-up (S2 Table).

In multivariable LME models, urinary concentrations of BPA and phthalate metabolite groups were not associated with eGFR, UPCR, SBP, or DBP (Table 3). However, PA was

**Table 1. Characteristics of study population by study visit: Chronic Kidney Disease in Children (CKiD) study, 2005–2015.**

| Characteristic | Baseline | Visit 1 | Visit 2 | Visit 3 | Visit 4 | Visit 5 |
|---|---|---|---|---|---|---|
| Number of participants | 538 | 523 | 470 | 416 | 295 | 227 |
| *Participant characteristics* | | | | | | |
| Age (years), median [IQR] | 10.98 [7.63, 14.56] | 11.37 [9.02, 14.92] | 12.25 [8.95, 15.77] | 12.94 [9.54, 16.71] | 13.54 [10.31, 17.23] | 14.02 [10.44, 17.61] |
| Time from baseline (months), mean (SD) | 0.00 (0.00) | 7.56 (2.57) | 19.71 (3.55) | 31.82 (3.95) | 44.31 (3.87) | 56.12 (3.85) |
| Sex = male, *N* (%) | 344 (63.9) | 327 (62.5) | 303 (64.5) | 268 (64.4) | 185 (62.7) | 139 (61.2) |
| Race/ethnicity, *N* (%) | | | | | | |
| Hispanic | 76 (14.1) | 74 (14.1) | 64 (13.6) | 57 (13.7) | 38 (12.9) | 33 (14.5) |
| Non-Hispanic white | 313 (58.2) | 305 (58.3) | 288 (61.3) | 259 (62.3) | 181 (61.4) | 137 (60.4) |
| Non-Hispanic black | 86 (16.0) | 79 (15.1) | 69 (14.7) | 56 (13.5) | 43 (14.6) | 33 (14.5) |
| Multi-race/other | 63 (11.7) | 65 (12.4) | 49 (10.4) | 44 (10.6) | 33 (11.2) | 24 (10.6) |
| Birth weight (kilograms), mean (SD) | 3.08 (0.75) | 3.07 (0.73) | 3.05 (0.76) | 3.03 (0.76) | 3.09 (0.70) | 3.04 (0.74) |
| Prematurity, *N* (%) | | | | | | |
| No | 453 (84.2) | 438 (83.7) | 394 (83.8) | 350 (84.1) | 255 (86.4) | 195 (85.9) |
| Yes | 62 (11.5) | 63 (12.0) | 61 (13.0) | 54 (13.0) | 34 (11.5) | 27 (11.9) |
| Missing | 23 (4.3) | 22 (4.2) | 15 (3.2) | 12 (2.9) | 6 (2.0) | 5 (2.2) |
| ACEI = yes, *N* (%) | 241 (44.8) | 244 (46.7) | 213 (45.3) | 194 (46.6) | 133 (45.1) | 98 (43.2) |
| ARB = yes, *N* (%) | 47 (8.7) | 40 (7.6) | 32 (6.8) | 38 (9.1) | 28 (9.5) | 23 (10.1) |
| BMI *z*-score, mean (SD) | 0.41 (1.15) | 0.34 (1.17) | 0.36 (1.12) | 0.31 (1.15) | 0.28 (1.16) | 0.25 (1.14) |
| Urinary creatinine (mg/dl), mean (SD) | 66.08 (57.78) | 168.08 (83.85) | 129.15 (89.38) | 69.09 (58.44) | 82.15 (51.28) | 61.32 (31.48) |
| Non-glomerular disease, *N* (%) | 475 (88.3) | 464 (88.7) | 432 (91.9) | 389 (93.5) | 276 (93.6) | 216 (95.2) |
| *Outcomes* | | | | | | |
| eGFR (ml/min/1.73 m$^2$), mean (SD) | 51.90 (19.80) | 51.47 (21.02) | 49.52 (20.62) | 48.74 (20.22) | 44.09 (19.48) | 41.10 (17.80) |
| UPCR (mg/dl:mg/dl), median [IQR] | 0.30 [0.11, 0.84] | 0.29 [0.12, 0.87] | 0.30 [0.11, 0.93] | 0.31 [0.12, 0.90] | 0.35 [0.14, 1.00] | 0.40 [0.11, 1.06] |
| SBP *z*-score, mean (SD) | 0.32 (1.08) | 0.34 (1.13) | 0.21 (1.21) | 0.14 (1.11) | 0.07 (1.07) | 0.18 (1.11) |
| DBP *z*-score, mean (SD) | 0.52 (0.99) | 0.41 (1.03) | 0.34 (0.98) | 0.27 (0.90) | 0.27 (0.98) | 0.30 (1.02) |
| 8-OHdG (ng/ml), median [IQR] | 30.02 [18.29, 48.11] | 23.68 [14.89, 38.13] | 23.92 [14.24, 43.36] | 32.62 [19.04, 51.77] | 38.28 [26.88, 54.60] | 33.04 [21.18, 49.96] |
| F$_2$-isoprostane (ng/ml), median [IQR] | 6.61 [0.88, 24.33] | 7.37 [0.95, 32.05] | 20.74 [8.08, 36.97] | 3.53 [0.71, 18.95] | 1.68 [0.64, 6.02] | 0.69 [0.43, 1.36] |
| NGAL (ng/ml), median [IQR] | 5.28 [1.30, 20.29] | 4.73 [0.93, 28.81] | 4.61 [0.72, 19.82] | 6.69 [1.60, 35.18] | 8.17 [2.18, 41.05] | 6.69 [2.14, 37.73] |
| KIM-1 (ng/ml), median [IQR] | 0.13 [0.03, 0.33] | 0.11 [0.03, 0.25] | 0.09 [0.01, 0.24] | 0.10 [0.02, 0.23] | 0.12 [0.07, 0.24] | 0.10 [0.04, 0.22] |
| *Exposures* | | | | | | |
| BPA (ng/ml), median [IQR] | 0.78 [0.18, 1.66] | 0.50 [0.26, 0.98] | 0.47 [0.27, 0.81] | 0.60 [0.11, 1.31] | 1.47 [0.40, 2.11] | 0.64 [0.11, 1.85] |
| PA (ng/ml), median [IQR] | 42.18 [14.75, 95.95] | 62.93 [36.93, 107.64] | 53.89 [30.00, 100.66] | 96.70 [55.01, 157.10] | 112.60 [78.12, 191.65] | 104.60 [62.35, 169.15] |
| LMW phthalates (nmol/ml), median [IQR] | 0.16 [0.09, 0.35] | 0.25 [0.12, 0.49] | 0.18 [0.10, 0.35] | 0.18 [0.10, 0.34] | 0.23 [0.13, 0.46] | 0.18 [0.11, 0.37] |
| HMW phthalates (nmol/ml), median [IQR] | 0.18 [0.10, 0.32] | 0.14 [0.08, 0.27] | 0.18 [0.11, 0.30] | 0.22 [0.13, 0.37] | 0.23 [0.15, 0.39] | 0.21 [0.13, 0.35] |
| DEHP (nmol/ml), median [IQR] | 0.08 [0.04, 0.17] | 0.05 [0.03, 0.11] | 0.10 [0.06, 0.18] | 0.11 [0.06, 0.22] | 0.10 [0.07, 0.18] | 0.09 [0.06, 0.15] |
| DOP (nmol/ml), median [IQR] | 0.02 [0.01, 0.05] | 0.01 [0.01, 0.03] | 0.01 [0.00, 0.02] | 0.01 [0.01, 0.01] | 0.01 [0.01, 0.02] | 0.01 [0.01, 0.02] |

Percent missing for covariates: age, sex, race/ethnicity, glomerular status, ARB/ACEI use, creatinine, 0%; birthweight, 5.5%; prematurity, 4.2%; BMI, 3.2%; SBP, 7.3%.
8-OHdG, 8-hydroxy-2′-deoxyguanosine; ACEI, angiotensin-converting enzyme inhibitor; ARB, angiotensin receptor blocker; BMI, body mass index; BPA, bisphenol A; DBP, diastolic blood pressure; DEHP, di(2-ethylhexyl) phthalate; DOP, di-n-octyl phthalate; eGFR, estimated glomerular filtration rate; HMW, high molecular weight; IQR, interquartile range; KIM-1, kidney injury molecule-1; LMW, low molecular weight; NGAL, neutrophil gelatinase-associated lipocalin; PA, phthalic acid; SBP, systolic blood pressure; SD, standard deviation; UPCR, urinary protein-to-creatinine ratio.

**Table 2.  Distributions of urinary BPA, PA, and phthalate metabolites collected over time.**

| Chemical | LOD | Percent < LOD | GM | GSD | Percentile | | | |
|---|---|---|---|---|---|---|---|---|
| | | | | | 25th | 50th | 75th | 95th |
| Bisphenol A (BPA) (ng/ml) | 0.15 | 21.7 | 0.59 | 3.49 | 0.22 | 0.60 | 1.42 | 4.53 |
| Phthalic acid (PA) (ng/ml) | 0.122 | 0.9 | 63.95 | 3.43 | 35.80 | 72.83 | 129.35 | 355.26 |
| *Individual phthalate metabolites* | | | | | | | | |
| Monomethylphthalate (mMP) (ng/ml) | 0.027 | 6.2 | 1.59 | 4.43 | 1.00 | 1.92 | 3.59 | 10.54 |
| Monoethylphthalate (mEP) (ng/ml) | 0.035 | 0.2 | 21.43 | 3.78 | 8.86 | 18.44 | 44.87 | 223.12 |
| Mono(3-carboxypropyl) phthalate (mCPP) (ng/ml) | 0.221 | 2.8 | 2.22 | 2.80 | 1.22 | 2.14 | 3.88 | 12.26 |
| Mono(2-ethyl-5-hydroxyhexyl) phthalate (mEHHP) (ng/ml) | 0.035 | 0.0 | 5.44 | 3.32 | 2.43 | 5.52 | 11.60 | 41.72 |
| Monobenzyl phthalate (mBzP) (ng/ml) | 0.03 | 1.3 | 7.85 | 3.61 | 4.06 | 8.26 | 16.91 | 49.98 |
| Monocyclohexyl phthalate (mCHP) (ng/ml) | 0.038 | 83.9 | — | — | <LOD | <LOD | <LOD | 0.14 |
| Mono-n-octyl phthalate (mOP) (ng/ml) | 0.036 | 63.9 | — | — | <LOD | <LOD | 0.2 | 1.51 |
| Mono(8-methyl-1-nonyl) phthalate (mIDP) (ng/ml) | 0.016 | 63.8 | — | — | <LOD | <LOD | 0.44 | 3.82 |
| Mono-hexyl phthalate (mHxP) (ng/ml) | 0.018 | 36.2 | 0.09 | 5.72 | <LOD | 0.12 | 0.35 | 1.35 |
| Mono(2-heptyl) phthalate (mHPP) (ng/ml) | 0.029 | 30.8 | 0.26 | 7.42 | <LOD | 0.38 | 1.28 | 4.57 |
| Mono(carboxyisooctyl) phthalate (mCIOP) (ng/ml) | 0.15 | 38.4 | 1.17 | 8.98 | <LOD | 1.67 | 7.32 | 36.36 |
| Mono(carboxy-isononyl) phthalate (mCINP) (ng/ml) | 0.169 | 43.4 | 0.45 | 4.11 | <LOD | 0.32 | 1.39 | 5.22 |
| Mono-n-pentenyl phthalate (mPeP) (ng/ml) | 0.187 | 95.4 | — | — | <LOD | <LOD | <LOD | <LOD |
| Monoisopropyl phthalate (mIPrP) (ng/ml) | 0.146 | 75.1 | — | — | <LOD | <LOD | <LOD | 2.47 |
| Monobutyl phthalate (mBP) (ng/ml) | 0.039 | 0.1 | 8.92 | 2.59 | 4.91 | 8.91 | 15.77 | 41.56 |
| Mono-isobutyl phthalate (mIBP) (ng/ml) | 0.019 | 6.2 | 2.86 | 5.38 | 1.91 | 3.62 | 7.24 | 20.72 |
| Mono(2-ethyl-5-carboxypentyl) phthalate (mECPP) (ng/ml) | 0.046 | 0.5 | 6.8 | 4.55 | 3.09 | 9.06 | 18.10 | 59.26 |
| Mono(2-carboxymethylhexyl) phthalate (mCMHP) (ng/ml) | 0.173 | 14.4 | 3.06 | 5.50 | 1.47 | 4.12 | 9.48 | 33.23 |
| Mono(7-carboxy-n-heptyl) phthalate (mCHpP) (ng/ml) | 0.112 | 21.6 | 0.54 | 5.64 | 0.13 | 0.33 | 1.66 | 13.86 |
| Mono(2-ethyl-5-oxohexyl) phthalate (mEOHP) (ng/ml) | 0.016 | 8.7 | 2.48 | 8.20 | 1.16 | 3.64 | 8.64 | 41.00 |
| Monoisononyl phthalate (mINP) (ng/ml) | 0.018 | 63.4 | — | — | <LOD | <LOD | 0.17 | 2.57 |
| *Phthalate metabolite groups* | | | | | | | | |
| ΣLow-molecular weight (LMW) phthalates (nmol/ml) | — | — | 0.218 | 2.88 | 0.11 | 0.19 | 0.40 | 1.40 |
| ΣHigh-molecular weight (HMW) phthalates (nmol/ml) | — | — | 0.19 | 2.36 | 0.11 | 0.19 | 0.33 | 0.85 |
| ΣDi(2-ethylhexyl) phthalate (DEHP) (nmol/ml) | — | — | 0.09 | 2.64 | 0.05 | 0.09 | 0.17 | 0.54 |
| ΣDi-n-octyl phthalate (DOP) (nmol/ml) | — | — | 0.01 | 2.92 | 0.01 | 0.01 | 0.02 | 0.08 |

Geometric means were not calculated for chemicals with ≥50% of measures < LOD. ΣLMW phthalates is a micromolar sum of mMP, mEP, mBP, and mIBP. mCHP, mPeP, and mIPrP were excluded because they were detected in <50% of samples. ΣHMW phthalates is a micromolar sum of mCPP, mECPP, mEHHP, mEOHP, mBzP, mHxP, mHPP, mCMHP, mCHpP, mCIOP, and mCINP. mOP, mIDP, and mINP were excluded because they were detected in <50% of samples. ΣDEHP is a micromolar sum of mECPP, mEHHP, mEOHP, and mCMHP. ΣDOP is a micromolar sum of mCPP and mCHpP. mOP was excluded because it was detected in <50% of samples.

GM, geometric mean; GSD, geometric standard deviation; LOD, limit of detection.

associated with lower eGFR over time. For a 1-SD increase in ln-transformed PA, we estimated an average decrease in eGFR of 0.38 ml/min/1.73 m$^2$ (95% CI: −0.75, −0.01). PA was not associated with UPCR, SBP, or DBP.

In contrast to the findings for clinical renal outcomes, urinary concentrations of BPA, PA, and phthalate metabolite groups were associated with increased NGAL and KIM-1 over time (Table 4). For example, a 1-SD increase in ln-transformed BPA was associated with a 9.0% increase in NGAL (95% CI: 1.4%, 17.2%) and 11.7% increase in KIM-1 (95% CI: 5.6%, 18.2%) (calculated from Table 4 by computing e$^β$). The associations of PA, ΣHMW phthalates, and ΣDOP with KIM-1 varied over time and are shown in Fig 1. PA and ΣHMW phthalates were positively associated with KIM-1 and increased in magnitude through 3 years, but associations

**Table 3. Associations between ln-transformed chemical exposures and eGFR, ln-transformed UPCR, SBP z-score, and DBP z-score from linear mixed-effects models.**

| Outcome and exposure | β | 95% CI | p-Value |
|---|---|---|---|
| **eGFR[a] (N = 2,112)** | | | |
| BPA | −0.22 | −0.56, 0.13 | 0.22 |
| PA | −0.38 | −0.75, −0.01 | 0.04 |
| LMW phthalates | 0.23 | −0.16, 0.62 | 0.25 |
| HMW phthalates | 0.11 | −0.26, 0.48 | 0.55 |
| DEHP | 0.13 | −0.23, 0.49 | 0.49 |
| DOP | −0.30 | −0.68, 0.07 | 0.12 |
| **Ln-UPCR[a] (N = 2,052)** | | | |
| BPA | 0.01 | −0.02, 0.04 | 0.53 |
| PA | 0.02 | −0.02, 0.05 | 0.31 |
| LMW phthalates | −0.00 | −0.04, 0.04 | 0.90 |
| HMW phthalates | 0.01 | −0.02, 0.05 | 0.49 |
| DEHP | 0.02 | −0.01, 0.05 | 0.23 |
| DOP | 0.00 | −0.04, 0.04 | 0.98 |
| **SBP z-score[b] (N = 2,122)** | | | |
| BPA | 0.01 | −0.03, 0.05 | 0.66 |
| PA | 0.00 | −0.04, 0.04 | 0.94 |
| LMW phthalates | −0.01 | −0.05, 0.04 | 0.76 |
| HMW phthalates | 0.00 | −0.04, 0.04 | 0.95 |
| DEHP | 0.01 | −0.03, 0.05 | 0.55 |
| DOP | −0.01 | −0.05, 0.04 | 0.76 |
| **DBP z-score[b] (N = 2,121)** | | | |
| BPA | 0.01 | −0.03, 0.04 | 0.70 |
| PA | 0.02 | −0.02, 0.05 | 0.37 |
| LMW phthalates | 0.00 | −0.04, 0.04 | 0.99 |
| HMW phthalates | −0.01 | −0.05, 0.03 | 0.49 |
| DEHP | −0.01 | −0.05, 0.03 | 0.73 |
| DOP | −0.01 | −0.05, 0.03 | 0.60 |

β: estimate per SD change.

[a]The model controlled for age, sex, race/ethnicity, glomerular status, birth weight, prematurity, ARB, ACEI, BMI z-score, and systolic blood pressure z-score (all measured at each patient's first visit) and creatinine.

[b]The model controlled for age, sex, race/ethnicity, glomerular status, birth weight, prematurity, ARB, ACEI, and BMI z-score (all measured at each patient's first visit) and creatinine.

ACEI, angiotensin-converting enzyme inhibitor; ARB, angiotensin receptor blocker; BPA, bisphenol A; DBP, diastolic blood pressure; DEHP, di(2-ethylhexyl) phthalate; DOP, di-n-octyl phthalate; eGFR, estimated glomerular filtration rate; HMW, high molecular weight; LMW, low molecular weight; PA, phthalic acid; UPCR, urinary protein-to-creatinine ratio.

attenuated at year 4 and were null by year 5 (Fig 1; S3 Table). In contrast, the association of KIM-1 with ΣDOP was strongest at baseline and attenuated over time. Associations with NGAL were all positive and did not vary over time.

In sensitivity analyses, chemical exposures were considered cumulatively over time as time-weighted averages (S5–S9 Tables). Results were largely consistent, although the association between PA and eGFR was no longer statistically significant (β = −0.31; 95% CI: −0.92, 0.30), and there was a significant positive association between ΣLMW phthalates and eGFR (β = 1.56; 95% CI: 0.66, 2.47) (S4 Table). In addition, positive associations remained between

**Table 4. Associations between ln-transformed chemical exposures and ln-transformed tubular injury biomarkers from linear mixed-effects models.**

| Outcome and exposure | β | 95% CI | p-Value |
|---|---|---|---|
| **ln-NGAL (N = 2,010)** | | | |
| BPA | 0.09 | 0.01, 0.16 | 0.02 |
| PA | 0.20 | 0.13, 0.28 | <0.001 |
| LMW phthalates | 0.12 | 0.04, 0.20 | 0.004 |
| HMW phthalates | 0.14 | 0.06, 0.21 | <0.001 |
| DEHP | 0.11 | 0.04, 0.19 | 0.002 |
| DOP | 0.13 | 0.05, 0.21 | 0.001 |
| **ln-KIM-1 (N = 2,010)** | | | |
| BPA | 0.11 | 0.05, 0.17 | <0.001 |
| PA* | −0.01 | −0.08, 0.07 | 0.87 |
| LMW phthalates | 0.21 | 0.15, 0.27 | <0.001 |
| HMW phthalates* | 0.13 | 0.03, 0.22 | 0.01 |
| DEHP | 0.13 | 0.08, 0.19 | <0.001 |
| DOP* | 0.30 | 0.21, 0.40 | <0.001 |

β: estimate per SD change. All models controlled for age, sex, race/ethnicity, glomerular status, birth weight, premature birth, angiotensin receptor blocker, angiotensin-converting enzyme inhibitor, BMI z-score, and systolic blood pressure z-score (all measured at each patient's first visit) and creatinine.

*Exposure has significant interaction with time ($p < 0.05$), and estimate for exposure at baseline is presented here; time-specific estimates are plotted in Fig 1.

BPA, bisphenol A; DEHP, di(2-ethylhexyl) phthalate; DOP, di-n-octyl phthalate; HMW, high molecular weight; KIM-1, kidney injury molecule-1; LMW, low molecular weight; NGAL, neutrophil gelatinase-associated lipocalin; PA, phthalic acid.

BPA, PA, and phthalate metabolite groups were also positively associated with oxidative stress biomarkers, particularly 8-OHdG (Table 5). In addition, PA, ΣLMW phthalates, and ΣDOP were associated with increases in $F_2$-isoprostane. Associations between BPA and PA and 8-OHdG, and between BPA and ΣDEHP and $F_2$-isoprostane, were shown to increase over the course of follow-up (Fig 2; S4 Table).

chemical exposures and NGAL and KIM-1 (S6 and S7 Tables) and between chemical exposures and 8-OHdG and $F_2$-isoprostane (S8 and S9 Tables). Model fit diagnostics were similar comparing these models with the primary analyses. Finally, additionally controlling for calendar year of study enrollment did not measurably change the results of the primary analysis.

## Discussion

In this longitudinal study of 618 children with CKD, urinary concentrations of BPA and phthalate metabolites were not associated with eGFR, UPCR, SBP, and DBP, but PA was inversely associated with eGFR. Despite the general paucity of associations with clinically relevant renal outcomes, urinary concentrations of chemical exposures were robustly associated with biomarkers of tubular injury, NGAL and KIM-1. Specifically, serially assessed urinary concentrations of BPA, PA, and phthalate metabolites were associated with increased NGAL and KIM-1 concentrations over time. Similarly, chemical exposure concentrations were associated with increases in oxidative stress biomarkers, 8-OHdG and $F_2$-isoprostane, which suggests a potential mechanism of action through which these chemical exposures may impart renal injury. These observations were consistent when chemical exposures were considered cumulatively over time.

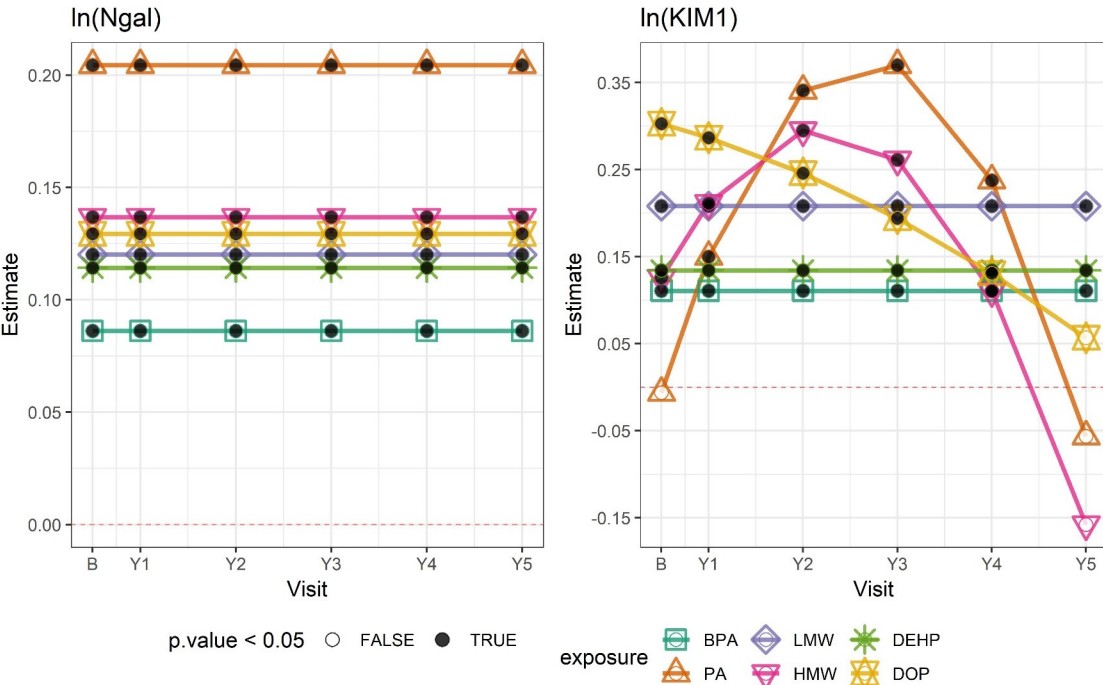

**Fig 1. Associations between ln-transformed chemical exposures and ln-transformed biomarkers of tubular injury by study visit.** Outcome estimates were derived from adjusted linear mixed-effects (LME) models and correspond to a 1–standard deviation change in each ln-transformed chemical exposure. Numerical estimates are shown in S3 Table. The horizontal dashed line indicates 0. Black dots indicate that the estimate had a *p*-value < 0.05; white dots indicate a *p*-value ≥ 0.05. B, baseline; BPA, bisphenol A; DEHP, di(2-ethylhexyl) phthalate; DOP, di-n-octyl phthalate; HMW, high molecular weight phthalates; KIM1, kidney injury molecule-1; LMW, low molecular weight phthalates; Ngal, neutrophil gelatinase-associated lipocalin; PA, phthalic acid; Y[number], year [number].

Our findings highlight the importance of considering environmental exposures as potentially modifiable risk factors for renal function decline and CKD progression. Although we did not detect associations of chemical exposures with clinical renal outcomes (i.e., eGFR), we report robust and consistent positive associations with biomarkers of tubular injury, NGAL and KIM-1. Although we observed some differences in these associations across time, these were mostly in magnitude and not direction, and the positive correlations between chemical exposures and NGAL and KIM-1 largely persisted. For example, the positive association of PA and ΣHMW phthalates with KIM-1 attenuated at 4 years of follow-up and was null at year 5, likely due to sparse counts in the later years of follow-up. These findings are novel and have important implications for long-term renal function. Several studies have documented NGAL and/or KIM-1 as early prognostic biomarkers of CKD and/or CKD progression [60–65], and one study identified KIM-1 as a cause of eGFR decline [61]. It is possible that we did not detect associations with clinical renal outcomes such as eGFR in this study because of the modest duration of follow-up, the relatively well-preserved eGFR over time, and the slow pace of progression of disease in children with CKD of tubular origin (i.e., congenital anomalies of the kidney and urinary tract). In addition, the incremental harm induced by environmental chemicals may be below the level of clinical detection, given the already existing underlying kidney disease of study participants, whereas subclinical impacts on tubular injury biomarkers may have been an easier signal to identify. This proposal is supported by our findings of increased oxidative stress in relation to BPA and phthalates. Given the literature supporting strong associations between both tubular injury and oxidative stress biomarkers and CKD progression

**Table 5. Associations between ln-transformed chemical exposures and ln-transformed oxidative stress biomarkers from linear mixed-effects models.**

| Outcome and exposure | β | 95% CI | p-Value |
|---|---|---|---|
| **8-OHdG (N = 2,118)** | | | |
| BPA* | 0.05 | −0.00, 0.10 | 0.06 |
| PA* | 0.10 | 0.05, 0.14 | <0.001 |
| LMW phthalates | 0.17 | 0.14, 0.21 | <0.001 |
| HMW phthalates | 0.18 | 0.15, 0.21 | <0.001 |
| DEHP | 0.16 | 0.13, 0.19 | <0.001 |
| DOP | 0.10 | 0.06, 0.13 | <0.001 |
| **F$_2$-isoprostane (N = 1,113)** | | | |
| BPA* | 0.11 | −0.15, 0.37 | 0.40 |
| PA | 0.17 | 0.03, 0.31 | 0.02 |
| LMW phthalates | 0.19 | 0.08, 0.30 | 0.001 |
| HMW phthalates | 0.08 | −0.03, 0.19 | 0.17 |
| DEHP* | −0.10 | −0.35, 0.14 | 0.41 |
| DOP | 0.13 | 0.01, 0.25 | 0.04 |

β: estimate per SD change. All models controlled for age, sex, race/ethnicity, glomerular status, birth weight, premature birth, angiotensin receptor blocker, angiotensin-converting enzyme inhibitor, BMI z-score, and systolic blood pressure z-score (all measured at each patient's first visit) and creatinine.

*Exposure has significant interaction with time (p < 0.05), and estimate for exposure at baseline is presented here; time-specific estimates are plotted in Fig 2.

8-OHdG, 8-hydroxy-2′-deoxyguanosine; BPA, bisphenol A; DEHP, di(2-ethylhexyl) phthalate; DOP, di-n-octyl phthalate; HMW, high molecular weight; LMW, low molecular weight; PA, phthalic acid.

and/or renal function, the observed changes in NGAL and KIM-1 and 8-OHdG and F$_2$-isoprostane in relation to BPA and phthalates raise concerns about the potential for clinically significant alterations in chronic kidney function to emerge in the long term.

The current analysis complements our previous work in which we analyzed the cross-sectional relationships between baseline levels of BPA and phthalates and kidney function measures at CKiD enrollment [16]. That study found that ΣLMW phthalates were associated with lower proteinuria and greater eGFR, which suggests a more favorable renal profile. In contrast, another recent study among hypertensive adults examined serum BPA in relation to CKD progression and found that baseline BPA levels were associated with decrements in eGFR over time [66]. In the current longitudinal investigation, we did not find any significant associations between urinary BPA and phthalates and clinical renal outcomes, except for an association between PA and lower eGFR over time. These disparate findings, and the addition of the findings for biomarkers of tubular injury, underscore the importance of incorporating both baseline and serial assessments of exposure to short-lived organic environmental chemicals to clarify the long-term effects of these exposures on kidney function.

BPA and phthalates have consistently been shown to be related to increased oxidative stress. Both oxidative DNA damage and lipid peroxidation have been documented in association with exposure to these chemicals in diverse populations, such as adolescents and pregnant and postmenopausal women [22,23,67–69]. Toxicological studies support these associations [18–20,70–72]. This study, conducted among children with CKD, found similar results. Taken in conjunction with our results that showed links between BPA and phthalates and markers of tubular injury, as well as the other studies that have documented associations between BPA and phthalates and deterioration in other measures of renal function [15,16,73–75], this study

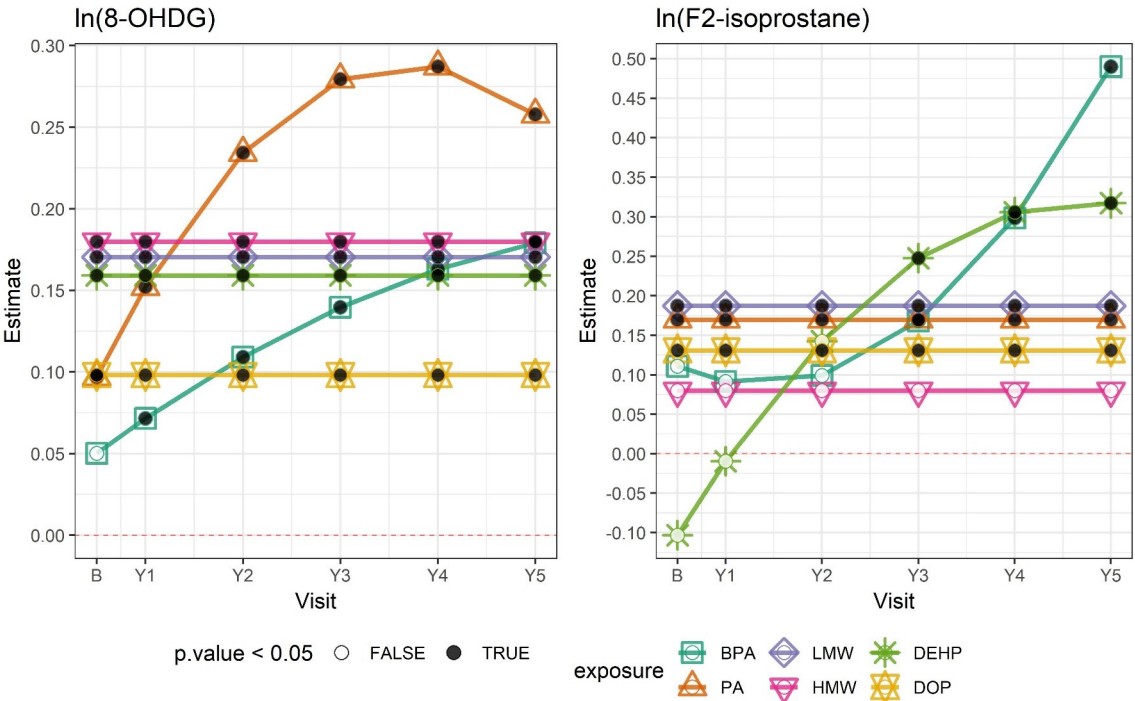

**Fig 2. Associations between ln-transformed chemical exposures and ln-transformed oxidative stress biomarkers by study visit.**
Outcome estimates were derived from adjusted linear mixed-effects (LME) models and correspond to a 1–standard deviation change
in each ln-transformed chemical exposure. Numerical estimates are shown in S4 Table. The horizontal dashed line indicates 0. Black
dots indicate that the estimate had a $p$-value < 0.05; white dots indicate a $p$-value ≥ 0.05. 8-OHDG: 8-hydroxy-2′-deoxyguanosine; B,
baseline; BPA, bisphenol A; DEHP, di(2-ethylhexyl) phthalate; DOP, di-n-octyl phthalate; HMW, high molecular weight phthalates;
LMW, low molecular weight phthalates; PA, phthalic acid; Y[number], year [number].

provides preliminary evidence for oxidative stress as a potential biological pathway for envi-
ronmental chemicals to influence kidney function.

This study benefited from several strengths. First, the longitudinal study design allowed us
to examine chemical exposures, renal function, and biomarkers of tubular injury and oxidative
stress over time. This was particularly important in this study of non-persistent chemical expo-
sures. These chemicals have been shown to be metabolized rapidly in the human body, with
half-lives ranging from 3 to 16 hours [43,44,46], and thus exhibit substantial intra-individual
variability in spot urine samples [76,77]. Furthermore, we employed a time-averaged statistical
approach, which allowed us to evaluate whether there were any associations with cumulative
levels of these exposures over time. Second, we measured several chemical exposures, as well as
diverse biomarkers of both tubular injury and oxidative stress, which complemented the clini-
cal data we had on renal function. The measures of tubular injury, NGAL and KIM-1, were of
paramount importance, in light of the lack of impact of the exposures on the clinically relevant
renal endpoints. In addition, measuring both NGAL and KIM-1 was important because
NGAL, normally present in the kidney but rapidly upregulated in response to renal injury, rep-
resents damage specifically to the distal tubule [78], while KIM-1, a type I cell membrane gly-
coprotein that is typically undetectable in the healthy kidney but is synthesized rapidly
following acute kidney injury, represents damage to the proximal tubule [79]. With regard to
oxidative stress, we had measures of both DNA damage (i.e., 8-OHdG) [80] and lipid peroxi-
dation (i.e., $F_2$-isoprostane) [81], which allowed us to gain a more comprehensive picture of
the chemical influences on different types of oxidative damage. Lastly, we were able to examine

a unique cohort of children with CKD from across the US and Canada that had detailed longitudinal clinical information, including eGFR and UPCR.

However, there were some limitations. Spot urine samples were used for exposure assessment, and BPA and phthalates, the targets of interest in this study, are non-persistent and rapidly metabolized in the body. Indeed, the exposures in this study had a large degree of within-person variability and poor reliability over time, which is similar to other studies of BPA and phthalates [47,77]. However, our longitudinal design and cumulative average exposure approach attempted to ameliorate this issue and represent an improvement over previous studies. Furthermore, measuring urinary analytes among those with kidney dysfunction is complex because the kidneys are responsible for filtering bodily waste products that are ultimately excreted [39]. If that process is impaired, the concentration of analytes in urine may in turn be affected. Since the exposures and many of the outcomes of interest in this study were measured in the urine, this could potentially impact study results. Another potential limitation is that we did not have information on the sources of environmental exposure, which could help translate these findings into actionable prevention efforts. We measured exposure to each of the chemicals by using biomarkers, which represent not only exposures from multiple sources, but also differences in metabolism of these exposures. For example, phthalate exposure could result from food packaging, personal care products, and medical devices or pharmaceuticals; the last of these likely has increased importance in this population of children with CKD [33–36]. In addition, if we had had information on potential sources of exposure, we could have evaluated whether the amount excreted in urine was correlated with estimated exposure dose, a particular concern among those with kidney disease or impaired excretory capacity. Another limitation was that even though we had longitudinal data over time, our sample size was not large. Lastly, as is the case with all observational studies, our study may have been vulnerable to unmeasured confounding.

In conclusion, in a longitudinal study among children with CKD, we documented consistent and robust associations between serially assessed urinary concentrations of common environmental toxicants, BPA and phthalates, and increased urinary levels of NGAL and KIM-1 over time, reflecting tubular injury in the kidney. However, it is important to note that we did not detect consistent associations between these chemicals and the longitudinal trajectory of clinically relevant renal outcomes, including eGFR, proteinuria, and blood pressure. Finally, we found that BPA and phthalates were associated with increased oxidative stress, which provides a potential pathway through which these chemical exposures may induce damage to the kidney. This detected signal of increased tubular injury and oxidative stress in relation to environmental exposures at current levels raises concern about the potential for clinically relevant changes to renal function to develop over time. Given that this study was conducted among children, these findings highlight the importance of considering environmental exposures in the context of CKD management before clinical changes are observed. Furthermore, attention to the impact of exposure to environmental chemicals on the course of CKD in children is important because it is a potentially modifiable risk factor.

## Supporting information

**S1 STROBE Checklist. STROBE, Strengthening the Reporting of Observational Studies in Epidemiology.**
(DOC)

**S1 Fig. Modeling strategy for linear mixed-effects models with random intercept and AR (1) error term.**
(DOCX)

**S1 Table. Intraclass correlation coefficients (ICCs) and 95% confidence intervals (CIs) for ln-transformed chemical exposures and outcomes over time.**
(DOCX)

**S2 Table. Associations between calendar year and urinary chemical concentrations over time from linear mixed-effects models.**
(DOCX)

**S3 Table. Time-specific estimates for the associations between ln-transformed chemical exposures and ln-transformed tubular injury biomarkers from linear mixed-effects models as shown in Fig 1.**
(DOCX)

**S4 Table. Time-specific estimates for the associations between ln-transformed chemical exposures and ln-transformed oxidative stress biomarkers from linear mixed-effects models as shown in Fig 2.**
(DOCX)

**S5 Table. Associations between cumulative average ln-transformed chemical exposures and eGFR, ln-transformed urinary protein-to-creatinine ratio, SBP *z*-score, and DBP *z*-score from linear mixed-effects models.**
(DOCX)

**S6 Table. Associations between cumulative average ln-transformed chemical exposures and ln-transformed kidney injury biomarkers from linear mixed-effects models.**
(DOCX)

**S7 Table. Time-specific estimates for associations between cumulative average ln-transformed chemical exposures and ln-transformed biomarkers of tubular injury from linear mixed-effects models.**
(DOCX)

**S8 Table. Associations between cumulative average ln-transformed chemical exposures and ln-transformed oxidative stress biomarkers from linear mixed-effects models.**
(DOCX)

**S9 Table. Time-specific estimates for associations between cumulative average ln-transformed chemical exposures and ln-transformed oxidative stress biomarkers from linear mixed-effects models.**
(DOCX)

**S1 Text. Analysis plan.**
(DOCX)

## Author Contributions

**Conceptualization:** Mengling Liu, Rajendiran Karthikraj, Kurunthachalam Kannan, Bradley A. Warady, Susan Furth, Howard Trachtman, Leonardo Trasande.

**Data curation:** Yinxiang Wu, Mengling Liu, Teresa M. Attina, Rajendiran Karthikraj, Kurunthachalam Kannan, Bradley A. Warady, Susan Furth, Howard Trachtman, Leonardo Trasande.

**Formal analysis:** Melanie H. Jacobson, Yinxiang Wu, Mengling Liu, Teresa M. Attina, Rajendiran Karthikraj, Kurunthachalam Kannan.

**Funding acquisition:** Mengling Liu, Mrudula Naidu, Kurunthachalam Kannan, Suzanne Vento, Howard Trachtman, Leonardo Trasande.

**Investigation:** Melanie H. Jacobson, Mengling Liu, Mrudula Naidu, Susan Furth, Suzanne Vento, Howard Trachtman, Leonardo Trasande.

**Methodology:** Melanie H. Jacobson, Mengling Liu, Howard Trachtman, Leonardo Trasande.

**Project administration:** Mengling Liu, Mrudula Naidu, Bradley A. Warady, Susan Furth, Suzanne Vento, Howard Trachtman, Leonardo Trasande.

**Software:** Yinxiang Wu.

**Supervision:** Mengling Liu, Howard Trachtman, Leonardo Trasande.

**Validation:** Melanie H. Jacobson, Howard Trachtman, Leonardo Trasande.

**Visualization:** Yinxiang Wu, Leonardo Trasande.

**Writing – original draft:** Melanie H. Jacobson.

**Writing – review & editing:** Melanie H. Jacobson, Yinxiang Wu, Howard Trachtman, Leonardo Trasande.

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
