## [Editor Report · Decision Letter 0]

18 Jan 2020

Dear Dr Jacobson, 

Thank you for submitting your manuscript entitled "Serially assessed bisphenol A and phthalate exposure is associated with increased tubular injury but not overt kidney function over time in children with chronic kidney disease" for consideration by PLOS Medicine.

Your manuscript has now been evaluated by the PLOS Medicine editorial staff and I am writing to let you know that we would like to send your submission out for external peer review.

**Please be aware that, due to the voluntary nature of our reviewers and academic editors, manuscript assessment may be subject to delays during the holiday season. Thank you for your patience.**

Kind regards,

Louise Gaynor-Brook, MBBS PhD,

PLOS Medicine

---

## [Decision Letter · Decision Letter 1]

3 Jul 2020

Dear Dr. Jacobson,

Thank you very much for submitting your manuscript "Serially assessed bisphenol A and phthalate exposure is associated with increased tubular injury but not overt kidney function over time in children with chronic kidney disease" (PMEDICINE-D-20-00109R1) for consideration at PLOS Medicine and please accept my sincere apologies for the unusual delay in getting back to you about it. 

Your paper was evaluated by a senior editor and discussed among all the editors here. It was also discussed with an academic editor with relevant expertise, and sent to independent reviewers, including a statistical reviewer. I should say we were waiting for an outstanding report, but decided not to delay our decision further based on the advice we have received.The reviews are appended at the bottom of this email and any accompanying reviewer attachments can be seen via the link below:

[LINK]

In light of these reviews, I am afraid that we will not be able to accept the manuscript for publication in the journal in its current form, but we would like to consider a revised version that addresses the reviewers' and editors' comments. Obviously we cannot make any decision about publication until we have seen the revised manuscript and your response, and we plan to seek re-review by one or more of the reviewers. 

We expect to receive your revised manuscript by Jul 24 2020 11:59PM. Please email us (plosmedicine@plos.org) if you have any questions or concerns.

We look forward to receiving your revised manuscript. 

Sincerely,

Clare Stone

Acting Editor in Chief

PLOS Medicine

plosmedicine.org

Please revise your title according to PLOS Medicine's style. Your title must be nondeclarative and not a question. It should begin with main concept if possible. "Effect of" should be used only if causality can be inferred, i.e., for an RCT. Please place the study design ("A randomized controlled trial," "A retrospective study," "A modelling study," etc.) in the subtitle (ie, after a colon).

Abstract – Please add summary demographic information, including mean age and please ensure here and throughout that p values are provided for all quantifiable data and where 95% Cis are given. Please provide a brief outline of the study’s limitations as the final sentence of the ‘Methods and Findings’ section of the abstract. 

Data – you state that some restrictions apply, then provide a link for the data. Can you clarify what restrictions there are? Is all of the data used in your analysis in the given link?

References – in the main text please use square brackets instead if rounded.

Line 151- “several sites”, please for relevance provide some of the cities.

Was written consent provided (from parents / guardians)?

Please provide a call-out to the analysis plan at the start of the Methods section (S1text)

Please address the comments from the Academic Editor below as well as the referee's comments. 

Comments from an Academic Editor:

The findings should be interpreted in the context of the fact that no associations were observed between the exposures and clinically relevant outcomes including eGFR (identified as the primary outcome), UPCR, SBP or DBP. Urinary NGAL and KIM-1 are recognised as markers of tubular injury but have not been shown to add additional prognostic value to traditional markers like eGFR and UPCR. In addition, the effect size was modest such that a 1 SD increase in exposure was associated with only a 9% increase in urinary NGAL or a 12% increase in urinary KIM-1.

A major limitation of the study is that assessment of chemical exposure was based on assays performed on random spot urine samples. As mentioned by the authors, the half life of these chemical is short (3-16 hours) so it is not clear whether random urine values accurately reflect 24 excretion.

Comments from the reviewers:

Reviewer #1: Statistical review

This study is a cohort study examining association between exposure to bisphenol A and phthalates on various measures of kidney function and damage amongst children with chronic kidney disease.

I have some comments on the statistical methods and reporting, which are listed below. 

1. Abstract - just for clarity it should be clear how many separate phthalate metabolites were tested for association.

2. Abstract - if possible within space constraints, report the association between phthalic acid and eGFR over time as is done for the other associations above.

3. Were the groupings of phthalates pre-specified?

4. Page 10 - what is the interpretation of the intra- and inter-assay CVs? They seem broadly within the rules of thumb of being below 10 and 15 respectively, but not all. What effect would higher CVs have on the analysis?

5. Page 11 - I think for some cases it would be useful to explore how well the model fits the data. The authors do provide a sensitivity analysis to explore how cumulative effects of the exposures affect the clinical variables. When the results differed from the two approaches, perhaps the AIC could be used to show which model fitted better? 

6. Page 11 - was there much missing data, and how was it handled? Were there many patients who developed ESKD, and how were they accounted for in the analysis?

7. Methods/results - for biomarkers such as NGAL and KIM-1 where multiple exposures were associated, did the authors consider models with more than one exposure included? I would recommend this might be useful if the exposures are correlated, in order to determine whether they were independent associations or not.

James Wason

Reviewer #2: see attachment

[LINK]

---

## [Decision Letter · Decision Letter 2]

19 Aug 2020

Dear Dr. Jacobson,

Thank you very much for re-submitting your manuscript "Serially assessed bisphenol A and phthalate exposure and kidney function in children with chronic kidney disease: a longitudinal cohort study" (PMEDICINE-D-20-00109R2) for review by PLOS Medicine.

I have discussed the paper with my colleagues and the academic editor and it was also seen again by one reviewer. I am pleased to say that provided the remaining editorial and production issues are dealt with we are planning to accept the paper for publication in the journal.

[LINK]

We look forward to receiving the revised manuscript by Aug 26 2020 11:59PM. 

Sincerely,

Artur Arikainen

Associate Editor 

PLOS Medicine

plosmedicine.org

Requests from Editors:

1. Title: Please update to “Serially-assessed bisphenol A or phthalate exposure and association with kidney function in children with chronic kidney disease in the US and Canada: a longitudinal cohort study”

2. Short Title: Please update to “BPA and phthalates associations with tubular injury in children with CKD”

3. Lines 32-36: Please remove the funding and competing interests statements – these should be entered in the online submission form.

4. Data Availability Statement: Please include this: “The data are owned by a third party and the authors do not have permission to share the data. This study represents the work of an ancillary study to the parent CKiD study. Access to the data may be arranged through contacting the parent study through the given link.”

5. Abstract:

a. Please ensure that symbols are correctly displayed here and throughout the text, eg. line 57: “…Σdi-n-octyl phthalate metabolites…” (first symbol appears as a large sigma)

b. Please mention any factors adjusted for in your analyses.

c. Please use square brackets when nesting, eg.: “(…[…]…)”

d. Please quantify all results with 95% CIs and p values (eg. lines 58-59), including statistically non-significant findings mentioned here (eg. lines 60-61).

e. Please remove this sentence (lines 70-73): "This is the first study to examine environmental exposures and biomarkers of tubular injury and oxidant stress in a pediatric population with CKD, and may provide evidence for oxidant stress as a plausible biologic pathway for environmental chemicals to influence kidney function.”

6. Line 87: Please correct to “…bisphenol A, phthalates and biomarkers…”

7. Line 93: Please correct to "…neither bisphenol A nor other..."

8. Please remove spaces from within citation callouts, eg. “…prenatally and in early life [15,26],…”, but keep the space between the text and the callout itself as is.

9. Methods:

a. Please add the following statement, or similar, to the Methods: "This study is reported as per the Strengthening the Reporting of Observational Studies in Epidemiology (STROBE) guideline (S1 Checklist)." 

b. Line 193: Please provide exact sample collection dates, including month and day.

c. Line 198: Please clarify whether parental consent was written or oral.

d. Line 250: Please remove trademark symbols.

e. Line 299 (and 423 in the main Discussion): Please avoid causal language: “effects”; please replace with “associations”.

10. Line 357: PLOS does not permit "data not shown.” Please remove this claim, or do one of the following:

a) If you are the owner of the data relevant to this claim, please provide the data in accordance with the PLOS data policy, and update your Data Availability Statement as needed.

b) If the data not shown refer to a study from another group that has not been published, please cite personal communication in your manuscript text (it should not be included in the reference section). Please provide the name of the individual, the affiliation, and date of communication. The individual must provide PLOS Medicine written permission to be named for this purpose.

c) For any other circumstance, please contact the journal office ASAP.

11. At line 361, should that be "inversely associated"?

12. In the limitations paragraph in the Discussion (line 437 onwards), please mention the limited study size and the possibility of unmeasured confounding.

13. Line 443: Please delete extra comma.

14. Please provide full access details (eg. DOI or URL) for references 5, 58, and 59. Please also give author names for reference 1, and journal names for references 19-21. Please double check that all references are complete and accurate.

15. Table 4 and throughout, please report “p<0.0001” as “p<0.001” instead.

-----------

Comments from Reviewers:

Reviewer #1: Thank you to the authors for addressing my previous comments well. I have no further issues to raise.

[LINK]

---

## [Editor Report · Decision Letter 3]

11 Sep 2020

Dear Dr. Jacobson, 

On behalf of my colleagues and the academic editor, Dr. Maarten Taal, I am delighted to inform you that your manuscript entitled "Serially assessed bisphenol A or phthalate exposure and association with kidney function in children with chronic kidney disease in the US and Canada: a longitudinal cohort study" (PMEDICINE-D-20-00109R3) has been accepted for publication in PLOS Medicine. 

PRODUCTION PROCESS

PRESS

PROFILE INFORMATION

Thank you again for submitting the manuscript to PLOS Medicine. We look forward to publishing it. 

Best wishes, 

Maarten Taal, 

Renal Medicine 

PLOS Medicine

plosmedicine.org